# Effect of Audiovisual Cross-Modal Conflict during Working Memory Tasks: A Near-Infrared Spectroscopy Study

**DOI:** 10.3390/brainsci12030349

**Published:** 2022-03-03

**Authors:** Jiahong Cui, Daisuke Sawamura, Satoshi Sakuraba, Ryuji Saito, Yoshinobu Tanabe, Hiroshi Miura, Masaaki Sugi, Kazuki Yoshida, Akihiro Watanabe, Yukina Tokikuni, Susumu Yoshida, Shinya Sakai

**Affiliations:** 1Graduate School of Health Sciences, Hokkaido University, Sapporo 060-0812, Japan; cjh612@outlook.com (J.C.); ryj.saito0217@gmail.com (R.S.); hiroshi_miura@houseikai.or.jp (H.M.); akki0226@eis.hokudai.ac.jp (A.W.); htj0812@eis.hokudai.ac.jp (Y.T.); 2Department of Rehabilitation Science, Faculty of Health Sciences, Hokkaido University, Sapporo 060-0812, Japan; ot-k-yoshida@huhp.hokudai.ac.jp (K.Y.); sakai@hs.hokudai.ac.jp (S.S.); 3Department of Rehabilitation Sciences, Health Sciences University of Hokkaido, Sapporo 061-0293, Japan; s-saku@hoku-iryo-u.ac.jp (S.S.); ysdssm@hoku-iryo-u.ac.jp (S.Y.); 4Department of Rehabilitation, Shinsapporo Paulo Hospital, Sapporo 004-0002, Japan; rx78ot@yahoo.co.jp; 5Department of Rehabilitation, Tokeidai Memorial Hospital, Sapporo 060-0031, Japan; lsbjkmsstks@gmail.com

**Keywords:** cross-modal conflict, ventrolateral prefrontal cortex (VLPFC), inferior parietal cortex (IPC), functional near-infrared spectroscopy (fNIRS), Paced Auditory Serial Addition Test (PASAT), Paced Visual Serial Addition Test (PVSAT)

## Abstract

Cognitive conflict effects are well characterized within unimodality. However, little is known about cross-modal conflicts and their neural bases. This study characterizes the two types of visual and auditory cross-modal conflicts through working memory tasks and brain activities. The participants consisted of 31 healthy, right-handed, young male adults. The Paced Auditory Serial Addition Test (PASAT) and the Paced Visual Serial Addition Test (PVSAT) were performed under distractor and no distractor conditions. Distractor conditions comprised two conditions in which either the PASAT or PVSAT was the target task, and the other was used as a distractor stimulus. Additionally, oxygenated hemoglobin (Oxy-Hb) concentration changes in the frontoparietal regions were measured during tasks. The results showed significantly lower PASAT performance under distractor conditions than under no distractor conditions, but not in the PVSAT. Oxy-Hb changes in the bilateral ventrolateral prefrontal cortex (VLPFC) and inferior parietal cortex (IPC) significantly increased in the PASAT with distractor compared with no distractor conditions, but not in the PVSAT. Furthermore, there were significant positive correlations between Δtask performance accuracy and ΔOxy-Hb in the bilateral IPC only in the PASAT. Visual cross-modal conflict significantly impairs auditory task performance, and bilateral VLPFC and IPC are key regions in inhibiting visual cross-modal distractors.

## 1. Introduction

The sensory environment in daily life is complex, and we commonly receive information from multiple sources through multiple sensory modalities. Visual and auditory stimuli are two primary sensory modalities in our lives [1]. Different sensory channels perceive visual and auditory stimuli, and diverse sensory areas of the brain process them. When we engage in an activity, we generally focus on the necessary information and suppress unnecessary information. In the incongruent information from unimodal visual or auditory stimuli or cross-modal audiovisual stimuli, unnecessary task-irrelevant information (e.g., advertising pops up while browsing a website, making a phone call while driving) seriously affects the processing of necessary information [1,2]. To resolve this conflict, cognitive control plays an important role in enhancing the processing of task-relevant information and suppressing task-irrelevant information [3,4]. Cognitive control is the process by which goals or plans influence behavior that can inhibit automatic responses and influence working memory (WM) [4,5,6]. It connects the processing of incoming sensory input and ensures that the actions performed are appropriate for the current environment [6]. In the early integration stage of input information, the perception and characteristic analysis of input stimuli occur through bottom-up processing [7]. If the stimuli have numerous mismatches at this stage, cognitive conflicts are triggered [8]. Afterward, top-down processing from higher-level (semantic or visuospatial) representations resolves cognitive conflict in accordance with the current goals and relevant modalities. In the top-down processing of cognitive control, previous neuroimaging studies have suggested that a network of prefrontal and parietal brain regions provides preparatory top-down control over the sensory cortex to prioritize task-relevant processing [9,10].

A previous literature review on the cognitive control of distractor suppression has demonstrated that most studies focused on unimodal (visual or auditory) conflicts [1]. In unimodal visual studies that investigated the neural correlates of the effects of distractions, increased ventrolateral prefrontal cortex (VLPFC) activity was shown during visual tasks with distractor conditions compared to visual tasks with no distractor conditions [11,12]. Additionally, the inferior parietal cortex (IPC) plays an important role in filtering visual distractors [13]. In unimodal auditory studies, inhibition of auditory distraction, which disrupts WM performance, requires the activation of the lateral prefrontal cortex, especially the VLPFC [14,15]. Meanwhile, few neuroimaging studies have focused on cross-modal conflicts; in particular, only a few studies have focused on auditory targets with visual distractors [1]. In a cross-modal behavioral study, auditory processing was more impacted by visual distractors compared to the effect of auditory distractors on visual processing, and these were assessed using long reaction times and high error rates [16]. Regarding neuroimaging studies, some evidence suggests that similar neural mechanisms are recruited for unimodal and cross-modal interference control. Previous studies have indicated that the prefrontal and parietal cortices are also involved in cross-modal processing, except for some specific visual and auditory processing brain areas [17,18], while top-down modulation by prefrontal regions may involve direct crosstalk between sensory cortices. Meanwhile, during conditions of divided attention, increased bilateral prefrontal and left IPC activation have been found in cross-modal conditions compared to unimodal conditions [19]. Several asymmetry mechanisms in the cross-modal interference control of both visual and auditory modalities have been reported. Previous studies have suggested that, unlike visual information, cross-modal auditory distraction can be inhibited at very early stages (i.e., prior to cortical processing) [20,21]. An audiovisual cross-modal study using electroencephalography (EEG) found that, for both younger and older adults, there was no significant visual cross-modal suppression when attending to auditory tasks, while there was significant auditory cross-modal suppression during visual tasks [22]. Moreover, neuroimaging studies have revealed that higher cognitive load in a WM visual task increases the level of activity in the prefrontal and parietal cortices, including the VLPFC, IPC, and dorsolateral prefrontal cortex (DLPFC) [21], and the increase in the DLPFC was greater under higher WM load [23]. These findings suggest that the activation of the frontoparietal regions, VLPFC and IPC, are critical factors that inhibit cross-modal distractors, and DLPFC is associated with WM load. Additionally, a PET study revealed that the visual areas were activated by irrelevant visual stimulus when performing auditory tasks, regardless of the load of the auditory task [24], whereas an fMRI study reported that the level of activity in the auditory cortex was decreased by the irrelevant auditory stimuli when performing a WM visual task [25]. These results suggested that the visual distractors were easier to process and interfered with the main task than auditory distractors. Taken together, the aforementioned results suggest an asymmetrical behavioral performance and activation of the frontoparietal cortical region between visual cross-modal and auditory cross-modal conflicts. However, to the best of our knowledge, no study has directly examined asymmetrical behavioral performance and activation using a single WM task with either the visual or the auditory task as the target task and the other as the distractor.

Most previous studies have focused on either auditory stimulus conflicts with visual stimuli or visual stimulus conflicts with auditory stimuli using visual and auditory stimuli that are different from each other (e.g., face and landscape images for visual stimuli, and voice, music, and some noises for auditory stimuli) [17,22]. In this case, it is difficult to control the intensity of each stimulus and the difficulty of the task; furthermore, the cognitive processing of the cognitive conflicts evoked after the input of the two stimuli is expected to be different. However, when the same task is used, cognitive processing after stimulus perception is expected to be similar, although the processing at the time of stimulus input differs depending on the modality. Therefore, confirming cross-modal conflicts using stimuli of the same nature across modalities can minimize the effects of other factors (stimulus intensity, task difficulty, cognitive processing required to perform the task, and individual differences [task-specific strengths and weaknesses] in the task). This allows us to identify the core mechanism underlying cross-modal conflicts. Additionally, it has been reported that unlike the case of using target tasks and distractors of different nature, semantic competition between stimuli occurs when the same task is used, resulting in stronger cognitive conflict [26]. Therefore, cross-modal conflicts that cause semantic competition should be examined separately since they are expected to have different strengths of cognitive conflicts than those of previous findings. Only three studies have characterized the two different types of cross-modal conflict through a single WM task with either the visual or auditory task as the target task and the other as the distractor using only behavioral performance and not brain activity [27,28,29]. One study reported that visual distraction disrupted performance during an auditory task, but auditory distraction did not disrupt performance during a visual task [27], whereas the other two showed that both visual and auditory distractors did not disrupt performance [28,29].

The paced serial addition test (PSAT) is a WM task that allows alternating targets and distractors with the same task content. It imposes high cognitive demands on multiple cognitive domains, such as WM, information processing speed, calculating ability, etc [30]. There are two types of PSAT tasks: the Paced Auditory Serial Addition Test (PASAT), which is an auditory modality task, and the Paced Visual Serial Addition Test (PVSAT), which is a visual modality task. Different inter-stimulus intervals (ISIs) have been used to modulate task difficulty (typically 2s/digit). The participants are required to respond correctly during ISI. Behavioral studies have suggested that PVSAT shows slightly higher task performance than PASAT [31,32]. However, both tests are interchangeable as high correlations (r > 0.7, *p* < 0.001) between both versions, even when using different delays, have been demonstrated [33]. Regarding neuroimaging, PASAT and PVSAT are associated with the bilateral frontoparietal regions [33].

Functional near-infrared spectroscopy (fNIRS) is a non-invasive device designed to monitor cerebral hemodynamics, a method commonly used to assess cerebral activity [34]. fNIRS is a small and portable machine that is less vulnerable to head and body motion artifacts and can measure brain activity in an environment closer to daily life. Moreover, fNIRS is much quieter than fMRI and less affected by electrical or magnetic interference from auditory devices, such as hearing aids or loudspeakers. These devices are either contraindicated or produce significant artifacts in fMRI, EEG, and MEG. Therefore, fNIRS is an ideal imaging technique for auditory research and has been used in various studies on hearing [35,36].

The purpose of this study is to characterize the two types of cross-modal conflicts that cause semantic competition (focusing on vision while ignoring auditory distractors and focusing on audition while ignoring visual distractors) through a single WM task and its brain activity, and the correlation between changes in task performance and changes in brain activity induced by audiovisual cross-modal conflicts. Being able to show correlations between these parameters would lead to stronger evidence for brain regions that play an important role in the modulation of cross-modal conflict. In previous studies, an anatomical region of interest (ROI) approach was used to assess the activity of the frontoparietal brain region that is associated with cognitive control during cross-modal conflicts and included frontoparietal regions. The hypotheses of this study for task performance were 2-fold; PASAT and PVSAT with distractor showed significantly lower task performance than that without distractor and PASAT with distractor showed significantly lower performance than PVSAT with distractor. Regarding the fNIRS results, the hypotheses of this study were also 2-fold; brain activity in the DLPFC, VLPFC, and IPC during PASAT and PVSAT with distractor were significantly higher compared with PASAT and PVSAT without distractor, and DLPFC activity that is a close link with cognitive load could increase more in PASAT with distractor than that in PVSAT with distractor based on the aforementioned previous studies. Furthermore, we also hypothesize that correlations between changes in behavioral performance and changes in brain activity exist.

## 2. Materials and Methods

### 2.1. Participants

Thirty-one right-handed, healthy young male participants were enrolled. All participants were native Japanese speakers. The required sample size for this study was calculated using a priori power analysis using G*power and the sample size to achieve a 0.90 statistical power level based on the results of a previous study [27]. To be conservative, 15% was added considering the possibility of dropout and outliers, and the determined sample size was 31. Regarding gender differences in response to the WM network (i.e., female participants often have more activity in limbic and prefrontal structures, whereas male participants have more activity in parietal regions), only male participants were recruited to maintain homogeneity of the subject sample [37]. All participants were right-handed, as assessed using the Japanese version of the FLANDERS handedness questionnaire [38], had no history of neurological or psychiatric disorders, and had no visual, auditory, or cognitive impairment that could affect the completion of tasks.

### 2.2. Experimental Design

All participants performed the two WM tasks, PSATs (PASAT and PVSAT) tasks with and without distractors in four conditions, and their brain activity during the task was measured using fNIRS. After completing the task, the participants’ degree of sleepiness was assessed using the Stanford Sleepiness Scale (SSS) to ensure participants’ alertness during tasks. For task performance, the task accuracy of the PSATs, percentage of correct responses (%), was the dependent variable and task type (PASAT or PVSAT), and distractor (with or without distractor) as the within-subject factors were the independent variables. Similarly, to identify brain activities during the tasks, changes in Oxy-Hb concentration ([m(mol/L) × mm]) in the ROIs (i.e., DLPFC, VLPFC, and IPC, see fNIRS instrument for details) were set as the dependent variable and task type (PASAT or PVSAT), and distractors (with distractor or without distractor), as the within-subject factors, were set as independent variables.

### 2.3. Experimental Task

Visual and auditory versions of the PSAT (PASAT and PVSAT) were administered using DMDX display software (University of Arizona, Tuscon, AZ, USA) [39]. Visual and auditory stimuli were simultaneously presented on the computer screen and sound speaker, respectively. Before the experiment, we explained the procedures for the four task conditions to all participants (PASAT without distractor, PVSAT without distractor, PASAT with visual distractor (PVSAT), PVSAT with auditory distractor (PASAT)), and they practiced them, as illustrated in Figure 1. 30 s before the experiment, the participants sat quietly to allow for homeostatic adaptation to the fNIRS, and the baseline data were recorded. The experiment was conducted in a block design in which a 61.0 s task (PASAT/PVSAT) interleaved with 30.0 s rest. The participants completed 12 blocks comprising three repeated runs into four task conditions randomly, based on the randomized function of the DMDX. These four task conditions were determined by the 2 × 2 factorial combination of the task type (PASAT vs. PVSAT) and the distractor (with distractor vs. without distractor), as illustrated in Figure 2. In each block, a randomized single digit between 1 and 9 was presented. The participants were required to sum up each digit with the previously presented digit. Each stimulus was presented every 2.0 s, and 29 answers were required in each block.

In the PASAT, auditory stimuli were presented through a sound speaker. The stimulus sound was set at 70 dB. Each stimulus was presented for 500 ms, followed by a 1500 ms ISI. A fixation point was presented at the center of the screen, and participants were instructed to gaze at it during the PASAT. In the PVSAT, single digits were presented in the center of the screen (stimulus size was determined as 4 cm × 4 cm, visual angle 3.27°), and every digit was displayed for 500 ms followed by 1500 ms of a fixation point. In the PASAT with visual distractor (PVSAT) task, auditory stimuli were presented simultaneously with the visual distractor (PVSAT). In the PVSAT with auditory distractor (PASAT) task, visual stimuli were presented on the screen simultaneously with the auditory distractor (PASAT).

To ensure their gaze for the fixation and their perception of the visual target or visual distractor stimuli, the ratio of fixation gazing during the task was measured using an eye tracker device (X60, Tobii Technology, Sweden) in each participant through the entire fNIRS recording phase. The ratio of fixation gazing during the task was calculated as the percentage of time that the participants stayed within a sphere consisting of a visual angle of 7° from the center of the screen for the total task duration.

### 2.4. Functional Near-Infrared Spectroscopy Instrument

Changes in Oxy-Hb concentration were measured using a multichannel fNIRS optical topography system (LABNIRS, Shimadzu Corporation, Kyoto, Japan) with three wavelengths of near-infrared light (780, 805, and 830 nm). The sampling rate was 6.17 Hz. The fNIRS probes comprised 16 illuminating and 14 detecting probes arranged alternately with an inter-probe distance of 3 cm, resulting in 38 channels. The probes were positioned over the six brain regions, bilateral DLPFC, VLPFC, and IPC, based on previous studies [8,18,23,24,33]. The positions of the probes and channels are shown in Figure 3a,b. All fNIRS probe positions and representative scalp landmarks (Cz, Nz, Iz, AL, and AR) were digitized using a three-dimensional magnetic space digitizer (FASTRAK; Polhemus, Colchester, VT, USA). These coordinate data were registered into Montreal Neurological Institute (MNI) coordinates using the “coordinate-based system” function in NIRS_SPM. The anatomical location of each channel was determined according to the Talairach Daemon [40]. The anatomical labeling (Brodmann areas, Talairach Daemon), which was averaged for all participants, is listed for each channel in Table 1. Activated channels were grouped into six ROIs, including the bilateral VLPFC (left: channels 1, 3, 6, and 8 were averaged; right: channels 14, 17, 19, and 22 were averaged), the bilateral DLPFC (left: channels 2, 4, 5, 7, and 9 were averaged; right: channels 13, 15, 16, 18, and 21 were averaged), and the bilateral IPC (left: channels 25, 27, 28, and 30 were averaged; right: channels 33, 35, 36, and 38 were averaged) (Figure 3a). All channels included in the IPC exceeded 60% of the estimated probability in individual-level registration, and all channels included in the VLPFC and DLPFC exceeded 60% of the estimated probability in mean-level registration. The optical fNIRS data were analyzed according to the modified Beer-Lambert-Law to quantify changes in Oxy-Hb, deoxygenated hemoglobin, and total hemoglobin concentration [41]. Our analysis only performs the changes in Oxy-Hb concentration, which provides the most representative indication of brain activity [42].

In our study, the baseline period comprised the 6-s period before task onset, and the average Oxy-Hb value of the baseline period was set to zero. In addition, a bandpass filter was then applied between 0.01 Hz and 0.3 Hz. To avoid NIRS path length issues, the changes in Oxy-Hb concentration during the task were calculated as the difference from the baseline value [43]. Similar to some previous studies, task-related changes in Oxy-Hb concentration in each ROI were averaged for the period during the task, from 6 s to 61 s after task onset [44,45].

### 2.5. Experimental Procedure

The experimental procedure consisted of three phases: phase I (i.e., practice phase), phase II (i.e., NIRS recording phase), and phase III (i.e., SSS phase) (Figure 1). In the practice phase, the participants were seated on a chair in front of a 27-inch computer monitor (1920 × 1080 pixels) and gazed at a fixation on the center of the screen to reduce eye and head movements. The participants wore the NIRS head cap and were instructed to avoid head and body motion and deep breathing during the NIRS measurements. Subsequently, the eye tracker system was calibrated for each participant before performing the task, and all participants received instructions for performing the four task conditions and practiced them for 19.0 s, which comprised nine items in each condition, making a total of 76.0 s for four conditions. During the NIRS recording phase, NIRS measurements were continuously acquired throughout the fNIRS recording phase. The fNIRS recording phase consisted of 12 blocks for a total of 1092.0 s (three blocks for each condition), including the 30.0 s baseline period, and each block consisted of 61.0 s, 30 stimuli. After performing the task, the participants were required to assess their degree of sleepiness through the SSS. The SSS is a self-rating scale and took approximately one minute to complete.

### 2.6. Statistical Analysis

A 2 × 2 repeated-measurement analysis of variance (ANOVA) with the task type (two levels: PASAT and PVSAT) and the distractor (two levels: with distractor, without distractor) as within-subject factors were performed for the task performance (percentage of correct responses). A 2 × 2 × 2 repeated-measurement ANOVA with the task type (two levels: PASAT and PVSAT), distractor (two levels: with and without distractor), and hemisphere (two levels: left and right) as within-subject factors was performed to determine the changes in Oxy-Hb concentration in each ROI. Furthermore, to elucidate the relationship between the changes in the percentage of correct responses in the PASAT or PVSAT with distractors compared to without distractors (Δpercentage of correct responses) of PASAT and PVSAT and the changes in Oxy-Hb concentration in ROIs with distractors compared to without distractors (∆Oxy-Hb), Pearson correlation analysis was performed. The Δpercentage of correct responses and ∆Oxy-Hb were calculated by subtracting the value in the no-distractor condition from the value in the distractor condition. All statistical analyses were performed using SPSS (version 25.0; IBM Corp., Armonk, NY, USA), and the statistical significance level was set at 0.05.

## 3. Results

### 3.1. Demographic Data

Three participants were excluded from all data analyses because of excessive artifacts and device malfunction. The final sample comprised 28 young male participants (aged 20–27 years, mean age = 23.08 ± 1.91, mean score of FLANDERS handedness questionnaire = 9.39 ± 1.79). None of the participants encountered any difficulties in performing the task. The mean ratio of fixation gazing during the tasks was 90.87% (five participants were excluded due to the high variability of the ratio of fixation gazing over mean ± 3SD), which was above 90%, indicating that participants adequately complied with the instructions to fixate on the visual target or distractor [28]. At the end of the experiment, the mean score of the SSS was 1.82 ± 0.98, indicating that the awakening state of participants was good during the experiment.

### 3.2. Behavioral Results

The repeated measures ANOVA of task performance accuracy, with task type and distractor as within-participants factors, revealed a significant main effect of task type (F(1,27) = 82.748, mean square error (MSE) = 94.901, *p* < 0.001, η^2^_p_ = 0.754) and distractor (F(1,27) = 14.741, MSE = 6.420, *p* = 0.001, η^2^_p_ = 0.353) and a significant task type × distractor interaction (F(1,27) = 7.696, MSE = 9.697, *p* = 0.010, η^2^_p_ = 0.222) (Figure 4). The simple main effect of task type showed significant lower accuracy in PASAT compared with in PVSAT in the no distractor condition (PASAT, 80.69 ± 13.31%, PVSAT, 95.80 ± 4.85%, F(1,27) = 66.010, MSE = 48.451, *p* < 0.001, η^2^_p_ = 0.710), and in the distractor condition ((PASAT with distractor, 77.21 ± 13.67%, PVSAT with distractor, 95.59 ± 6.02%, F(1,27) = 84.230, MSE = 56.147, *p* < 0.001, η^2^_p_ = 0.757). Moreover, a significant simple main effect of the distractor was observed in the PASAT (with distractor, 77.21 ± 13.67%, without distractor, 80.69 ± 13.31%, F(1,27) = 16.936, MSE = 9.959, *p* < 0.001, η^2^_p_ = 0.385), but not in the PVSAT (with distractor 95.59 ± 6.02%, without distractor, 95.80 ± 4.85%, F(1,27) = 0.061, MSE = 6.157, *p* = 0.759, η^2^_p_ = 0.004). Since there was a significant difference between PASAT and PVSAT, a paired-sample t-test was used to identify the existence of the cross-modal interfering effect by comparing the difference between [PASAT with distractor–PASAT] and [PVSAT with distractor–PVSAT]. The results revealed a significantly lower subtracting value of PASAT than that of PVSAT (*t*(27 )= −2.925, *p* = 0.007, Cohen’s *d* = 0.856).

### 3.3. Brain Activity Results

Our results showed no significant difference between the two types of tasks in all ROIs (*p* > 0.05). The results of 2 × 2 × 2 ANOVAs for the oxygenated hemoglobin (Oxy-Hb) concentration changes ROI analysis are summarized in Table 2.

In the VLPFC, no significant main effect of the hemisphere was revealed (F(1,27) = 0.973, MSE=0.000, *p* = 0.333, η^2^_p_ = 0.035), whereas a significant main effect of the task type (F(1,27) = 6.387, MSE = 0.002, *p* = 0.018, η^2^_p_ = 0.191) and the distractor (F(1,27) = 10.525, MSE = 0.001, *p* = 0.003, η^2^_p_ = 0.280), and interaction effect of task type × distractor (F(1,27) = 4.643, MSE = 0.001, *p* = 0.040, η^2^_p_ = 0.147) was observed. The simple main effect of distractor revealed a significantly increased activity in the PASAT in the distractor condition compared to PASAT in the no distractor condition (PASAT with distractor mean, 0.0347 ± 0.0389 [m(mol/L) × mm], PASAT without distractor mean, 0.0157 ± 0.0432 [m(mol/L) × mm], t(27) = 2.804, *p* = 0.009, Cohen’s *d* = 0.46), and the simple main effect of task type revealed a significantly increased activity in the PASAT in the distractor condition compared to PVSAT in the distractor condition (PASAT with distractor mean, 0.0347 ± 0.0389 [m(mol/L) × mm], PVSAT with distractor mean, 0.0126 ± 0.0348 [m(mol/L) × mm], t(27) = 3.281, *p* = 0.003, Cohen’s *d* = 0.60) (Figure 5). Conversely, no significant difference in Oxy-Hb concentration between with and without distractors in PVSAT (t(27) = 0.535, *p* = 0.597, Cohen’s *d* = 0.04) were observed.

In the DLPFC, a 2 × 2 × 2 ANOVA revealed no significant main effect of task type (F(1,27) = 4.185, MSE = 0.001, *p* = 0.051, η^2^_p_ = 0.134), distractor (F(1,27) = 1.809, MSE = 0.001, *p* = 0.190, η^2^_p_ = 0.063), hemisphere (F(1,27) = 0.315, MSE = 0.002, *p* = 0.579, η^2^_p_ = 0.012), and the interaction effect of task type × distractor (F(1,27) = 1.698, MSE = 0.001, *p* = 0.204, η^2^_p_ = 0.059).

In the IPC, a 2 × 2 × 2 ANOVA revealed no significant main effect of task type (F(1,27) = 0.027, MSE = 0.000, *p* = 0.871, η^2^_p_ = 0.001) and hemisphere (F(1,27) = 1.750, MSE = 0.001, *p* = 0.221, η^2^_p_ = 0.055), whereas there was a significant main effect of the distractor (F(1,27) = 4.278, MSE = 0.000, *p* = 0.048, η^2^_p_ = 0.137) and interaction effect of task type × distractor (F(1,27) = 6.008, MSE = 0.000, *p* = 0.021, η^2^_p_ = 0.182). The simple main effect of distractor showed a significantly increased activity in PASAT (with distractor mean, 0.0062±0.0176 [m(mol/L) × mm], without distractor mean, −0.0042 ± 0.0290 [m(mol/L) × mm], t(27) = 2.338, *p* = 0.027, Cohen’s *d* = 0.41), while there was no significant difference in PVSAT (t(27) = 0.308, *p* = 0.761, Cohen’s *d* = 0.02). Moreover, the simple main effect of task type showed a significantly increased activity in the distractor condition (PASAT with distractor mean, 0.0062 ± 0.0176 [m(mol/L) × mm], PVSAT with distractor mean, 0.0001 ± 0.0165 [m(mol/L) × mm], t(27) = 3.087, *p* = 0.005, Cohen’s *d* = 0.36).

### 3.4. Correlation Analysis

Figure 6 shows an overview of the correlations between the Δpercentage of correct responses in the PASAT and PVSAT and their ΔOxy-Hb concentration. There were significant positive correlations between Δtask performance accuracy (percentage of correct responses) and ΔOxy-Hb in bilateral IPC in the PASAT (Figure 7). Conversely, no significant correlation in any other region was observed in the PASAT (all, *p* > 0.05). Furthermore, there was no significant correlation between Δtask performance accuracy and ΔOxy-Hb in any ROI in the PVSAT (all, *p* > 0.05). All analyses were presented as uncorrected *p* < 0.05.

## 4. Discussion

The present study examined audiovisual cross-modal conflicts that cause semantic competition during WM tasks through behavioral and brain activities. According to our hypothesis, the PASAT and PVSAT would be more difficult in distractor conditions than in no distractor conditions. The results of this study clearly demonstrate that task type and distractor effect are important variables in determining performance on the PSAT. Specifically, the task performance accuracy of the PASAT was consistently lower than that of the PVSAT in either distractor or no-distractor conditions. In contrast, the distractor effect significantly interferes with task performance in the PASAT but not in the PVSAT. Regarding brain activity, a significant distractor effect was observed in the PASAT, in line with our hypothesis; however, this was not observed in the PVSAT. The change in Oxy-Hb concentration in the bilateral VLPFC and IPC significantly increased in the PASAT in the distractor condition compared to the no distractor condition, similar to our hypothesis. However, the Oxy-Hb concentration change in the bilateral DLPFC did not increase. Additionally, no significant change in Oxy-Hb concentration was observed in all ROIs in the PVSAT with distractor compared to without distractor conditions against the hypothesis. In the distractor condition, significantly increased Oxy-Hb concentration changes in the PASAT were observed in VLPFC and IPC in line with some of our hypotheses. Additionally, correlations between task performance and brain activity were only found in the bilateral IPC in PASAT but not in PVSAT.

### 4.1. Task Performance

For cross-modal distractors, there are different findings between visual cross-modal suppression and auditory cross-modal suppression. In visual cross-modal suppression, visual distractors disrupt auditory task performance [16,25,27]. In auditory cross-modal suppression, auditory distractors disrupt visual task performance [46,47], but some do not [27,28,47]. These results suggest that there was a modality-specific vulnerability to the distractor, indicating that it is relatively easy for young adults to inhibit auditory cross-modal distractors compared to visual cross-modal distractors. The results of this study are consistent with those of previous studies [16,25,27,28,47].

Regarding task type, our results showed a significantly lower task performance accuracy in the PASAT than in the PVSAT, which is supported by the findings of previous studies [32,33]. Tombaugh et al. suggested that a ceiling effect occurred when the ISI was 2.4 s [32]. When the ISI became progressively shorter, the task performance accuracy decreased faster for auditory stimuli than visual stimuli. As described above, the results of this study showed that the task difficulty of the PVSAT was lower than that of the PASAT, and the PVSAT may have nearly reached a ceiling effect in the current study (i.e., accuracy > 90%). These results suggest that low task difficulty may be one reason for the lack of significant cross-modal interference effects in the PVSAT. However, a previous study that used a single WM task reported that even when both auditory and visual tasks nearly reached the ceiling effect, significantly decreased performance was observed only in visual cross-modal conflicts [27]. This suggests that the effect of task difficulty alone cannot fully explain the differences between audio and visual cross-modal interference effects. Another possible reason for the lack of cross-modal interference effects may involve modality differences in filtering. The filtering mechanism plays an important role in protecting WM from interfering with irrelevant information in unimodality [48,49]. In cross-modal situations, the different filtering mechanisms between the visual and auditory modalities may need to be considered. Specifically, visual distractors were filtered out only at more central (e.g., visual cortex) processing levels, while auditory distractors were filtered out at both central (e.g., auditory cortex) and more peripheral (e.g., cochlea) neurocognitive levels that may be more easily and earlier filtered [28,50]. Additionally, a strong modality bias that changes through the lifespan of a human can influence lower cross-modal interference effects. Previous studies have reported that adults are more likely to have visual dominance [8,51]. Considering these reasons, differences in interference effect during cross-modal conflicts would depend on the nature of the task, including the engaged sensory modality, the level of difficulty of the task, and the modality of the distractor.

### 4.2. Brain Activity

A significantly increased activation was observed in the bilateral VLPFC and IPC in the PASAT in the distractor condition than in the no distractor condition, but no significant difference was found in the bilateral DLPFC. In addition, no significant difference was observed in the PVSAT. These results suggest that the increased activity observed in the bilateral VLPFC and IPC is associated with better interference control for behavioral performance. Successful cognitive control involves enhancing relevant stimuli, while the suppression of irrelevant stimuli through top-down control originates from prefrontal and parietal regions [48]. Numerous previous studies have reported that the bilateral VLPFC and IPC and its network are involved in visual distractor inhibition. In terms of cross-modal, some neuroimaging studies have shown that the VLPFC is activated during the processing of incongruent auditory-visual stimuli [17,18]. In line with these human neuroimaging studies, an electrophysiological study also showed the existence of cells in the VLPFC of macaques that respond to stimulation in more than one modality to accomplish cross-modal integration and conflict resolution (i.e., incongruent faces vs. vocalizations) [52]. These results suggest that the increased activity observed in the bilateral VLPFC is related to better control in the interference effect, the suppression of irrelevant stimuli, caused by the visual cross-modal distractor during the WM task that aids related cognitive control. Similar to the VLPFC, bilateral IPC has also been reported to play a crucial role in inhibiting distractors in visual unimodal studies [9,10,13]; however, there are insufficient reports on cross-modal studies. Vohn et al. [19] reported that left IPC significantly increased activity in the cross-modal tasks compared to unimodal tasks. Previous studies using brain stimulation (transcranial magnetic stimulation; TMS) demonstrated that inhibiting either the left or right IPC activity for visual distractors induces worth task performance [53,54]. In addition, a study of patients with bilateral parietal lesions also reported impaired performance in inhibiting visual distractors [55]. Furthermore, these studies argue that IPC applied an inhibition signal to the occipital region, which reduced the processing of salient distracting stimuli. Therefore, it is a common view in previous studies that these two regions play an important role in distractor inhibition. However, some evidence suggests differences in the contributions of distractor inhibition in the VLPFC and IPC, but this is still debatable. The inhibition of distractors is closely linked to the VLPFC [56,57], and WM storage is mainly associated with IPC [58,59]. Moreover, prefrontal regions play an important role in top-down control processing that supports active information storage in the IPC, which contributes to inhibiting distraction from sensory representations [11,12,49,60]. Therefore, the increased activity in the bilateral IPC may reflect the greater accumulation of unnecessary visual information due to stronger visual stimuli, as greater effort is required to inhibit visual distractors compared to auditory distractors.

Regarding the differences in distractor interference effect between the two tasks, the Oxy-Hb concentration changes in all six regions were significantly higher in PASAT than in PVSAT in the distractor condition. The bilateral VLPFC and DLPFC were examined in association with the central executive of the WM [11]. Regarding the conflict of audiovisual stimuli in cross-modal and unimodal conditions, previous neuroimaging studies have reported greater activity in the VLPFC, DLPFC, and IPC in unimodal conditions [11,12,14,26], whereas inconsistent results have been observed in cross-modal conditions [24,25,60,61,62]. These results may be associated with the nature of the sensory modality and cognitive load. Visual dominance over the auditory modality has been adequately demonstrated, as there is a higher behavioral cost and greater brain activity (i.e., the prefrontal cortex) when inhibiting a visual distractor [62]. Cross-modal auditory distraction is filtered at both central and more peripheral levels, which may occur at earlier stages of the interference control processing, while cross-modal visual distraction can be filtered primarily at the central level, which may occur at a later stage of the interference control processing and require higher levels of processing via top-down modulation accompanied with the activation of the frontoparietal regions [48]. Moreover, activation of the DLPFC is the critical factor underlying the failure to inhibit distractors; however, it does not occur in low-difficulty tasks [24,60,61]. Visual cross-modal distractors may always be processed in the visual cortex regardless of the load of the auditory task [25]. Therefore, visual cross-modal conflicts are expected to have a higher cognitive load than auditory cross-modal conflicts and require more DLPFC, VLPFC, and IPC activity. Specifically, these results may demonstrate that inhibition of visual cross-modal distractors requires more activation of the VLPFC and IPC, and failure to inhibit distractors induces increased activity in the DLPFC.

### 4.3. Correlations between Task Performance and Brain Activity

Correlation analyses revealed a significant positive correlation between ΔOxy-Hb concentration in the bilateral IPC and Δtask performance accuracy in the PASAT. This suggests that increased IPC activity contributes to decreasing the interference effect of visual cross-modal conflict for the target auditory task. The bilateral IPC has been identified to play an important role in WM processing and visual distractor inhibition by modulating the distraction from the visual processing area [11,12,19,49,60]. Moreover, the bilateral IPC is the absence of hemispheric specialization for inhibiting distractors [58,59]. However, no significant relationship was found in PVSAT. The low task difficulty and nature of the sensory modality may have affected this result. In this study, a high percentage of correct responses were observed in either the no distractor or the distractor conditions, and the percentage of correct responses were similar in both conditions. Therefore, the PVSAT with distractors may not recruit additional brain activity to inhibit the distractor [24,61].

## 5. Conclusions

This study characterized visual and auditory cross-modal conflict by comparing it to the no-distractor condition and compared them directly. It demonstrated asymmetrical behavioral performance and brain activity between the two types of cross-modal conflicts and gave more insight into the neural basis of cross-modal conflicts. The visual cross-modal interfering effect for auditory tasks significantly impaired its task performance, but the auditory cross-modal interference effect for visual tasks did not. The changes in Oxy-Hb concentration in the bilateral VLPFC and IPL showed significantly increased activity in visual cross-modal conflict than in the no distractor condition and auditory cross-modal conflict. Additionally, ΔOxy-Hb concentration in the bilateral IPC correlated positively with Δtask performance accuracy in the PASAT. These results suggest that the bilateral VLPFC and IPC play a pivotal role in decreasing the interference effect of visual cross-modal distractors. Future research should examine how each brain region inhibits cross-modal distractors and what functional differences exist between the hemispheres in these brain regions using neuromodulation techniques. Besides, the inclusion of reaction time of PASAT may allow us to see the trade-offs relationship between accuracy and speed, which is useful to better understand the impact of cross-modal conflicts. Further work should also investigate how the impact of visual and auditory cross-modal interference effect changes with the difficulty of the target WM task and how semantic and non-semantic cross-modal conflict differ in their effects on the WM task, which will contribute to a better understanding of the impact of cross-modal conflict and neural mechanisms.

## Figures and Tables

**Figure 1 brainsci-12-00349-f001:**
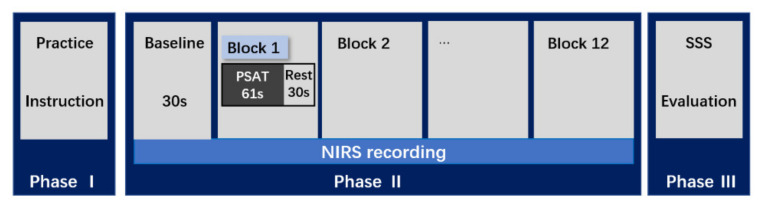
Experimental procedure. This study consisted of the following three phases: I: Practice phase; II: NIRS-recording phase; and III: SSS-evaluation phase. Phase II consisted of a 30 s baseline and twelve blocks, and each block consisted of 61 s PSAT task and 30-s rest. PSAT, Paced Serial Addition Test; fNIRS, functional near-infrared spectroscopy; SSS, Stanford Sleepiness Scale.

**Figure 2 brainsci-12-00349-f002:**
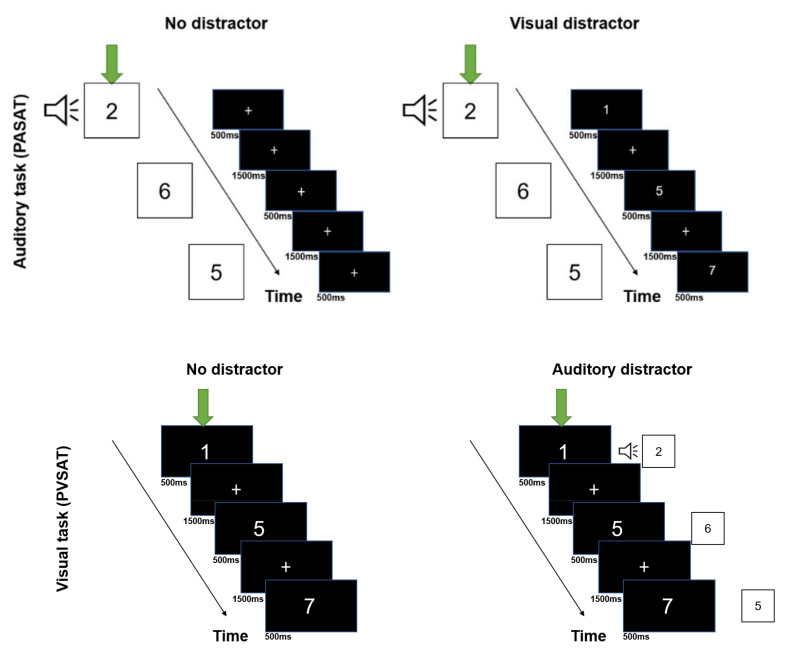
Experimental design. Experimental design of the Paced Auditory Serial Addition Test (top panel) and Paced Visual Serial Addition Test (bottom panel) without distractor (left side), with cross-modal distractors (right side). Green arrows indicate the target task.

**Figure 3 brainsci-12-00349-f003:**
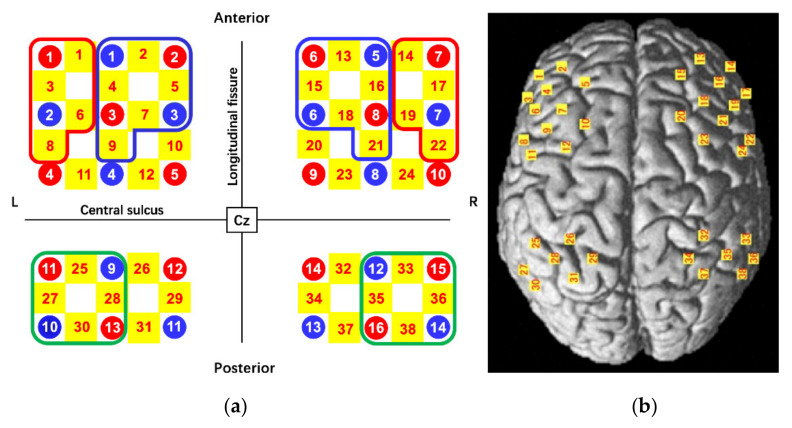
Near-infrared spectroscopy (NIRS) probe arrangement (Left anterior 3 × 3, posterior 3 × 2, Right anterior 3 × 3, posterior 3 × 2). (**a**) Illuminators are shown as red circles; detectors are shown as blue circles; channels are shown with a yellow-highlighted background. The Cz was defined according to the international 10–20 placement system. (**b**) The channel positions are shown on the cortical surface. Red, blue, and green frames show ventrolateral prefrontal cortex (VLPFC), dorsolateral prefrontal cortex (DLPFC) and Inferior parietal cortex (IPC), respectively.

**Figure 4 brainsci-12-00349-f004:**
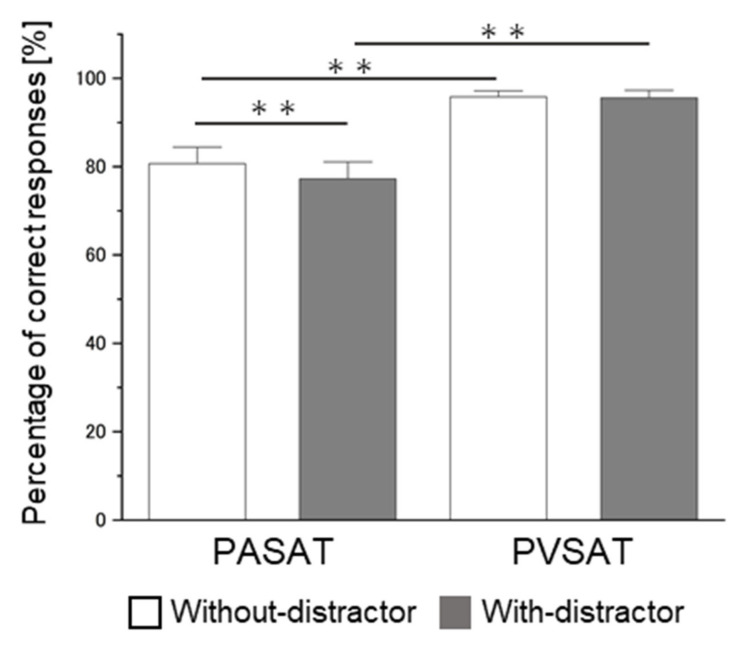
Results of the 2 × 2 analysis of variance of the percentage of correct responses (%) in the PSATs. Error bars indicate the standard error. PASAT, Paced Auditory Serial Addition Test; PVSAT, Paced Visual Serial Addition Test. ** *p* < 0.001.

**Figure 5 brainsci-12-00349-f005:**
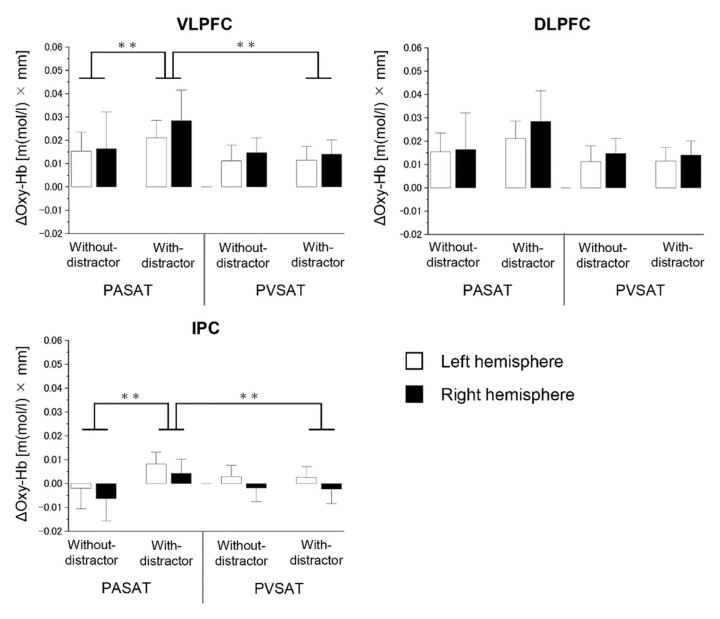
Results of the 2 × 2 × 2 analysis of variance of changes in Oxy-Hb concentration in each region of interest across PSATs task. The upper row shows the changes in Oxy-Hb concentration in the regions of interest (ROIs) on the VLPFC and DLPFC. The lower row shows the changes in Oxy-Hb concentration in the ROIs on the IPC. Error bars indicate the standard error. Oxy-Hb, oxygenated hemoglobin; VLPFC, ventrolateral prefrontal cortex; DLPFC, dorsolateral prefrontal cortex; IPC, Inferior parietal cortex; PASAT, Paced Auditory Serial Addition Test; PVSAT, Paced Visual Serial Addition Test. ** *p* < 0.01.

**Figure 6 brainsci-12-00349-f006:**
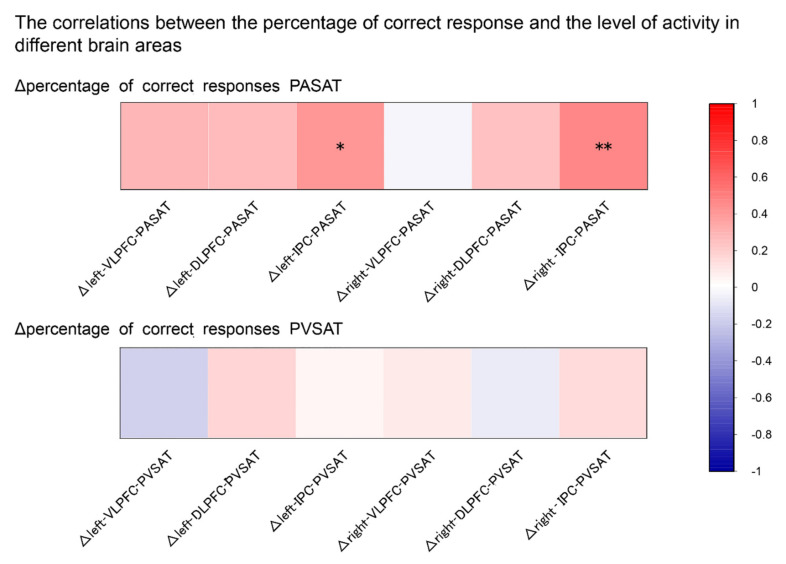
Correlation analyses between Δpercentage of correct responses in PASAT and PVSAT and ΔOxy-Hb concentration in all ROIs. The upper row indicates the correlations between Δpercentage of correct responses in the PASAT and ΔOxy-Hb concentration in all ROIs. The lower row indicates the correlations between Δpercentage of correct responses in the PVSAT and ΔOxy-Hb concentration in all ROIs. The color bar shows the correlation coefficient. VLPFC, ventrolateral prefrontal cortex; DLPFC, dorsolateral prefrontal cortex; IPC, Inferior parietal cortex; PASAT, Paced Auditory Serial Addition Test; PVSAT, Paced Visual Serial Addition Test. * *p* < 0.05, ** *p* < 0.01.

**Figure 7 brainsci-12-00349-f007:**
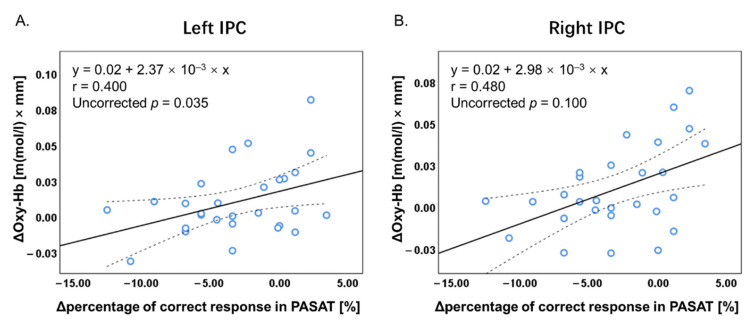
Scatter plots for Δpercentage of correct responses in PASAT and ΔOxy-Hb concentration in the bilateral IPC. (**A**) A significant positive correlation was found between Δpercentage of correct responses and ΔOxy-Hb concentration in left IPC in the PASAT. (**B**) A significant positive correlation was found between Δpercentage of correct responses and ΔOxy-Hb concentration in right IPC in the PASAT. Oxy-Hb, oxygenated hemoglobin; ROI, region of interest; IPC, inferior parietal cortex; PASAT, Paced Auditory Serial Addition Test.

**Table 1 brainsci-12-00349-t001:** Anatomical labeling of near-infrared spectroscopy (NIRS) channel positions.

Anatomical Labeling		Average Overlap Probability (%)	Channel Number	
Talairach Daemon	Brodmann Area		Left	Right
**Ventrolateral prefrontal cortex**	**44, 45, 47**	**71.57 (3.5)**	**1, 3, 6, 8**	**14, 17, 19, 22**
**Dorsolateral prefrontal cortex**	**9,46**	**70.78 (3.5)**	**2, 4, 5, 7, 9**	**13, 15, 16, 18, 21**
Includes Frontal eye field	8	65.18 (3.8)	10	20
Pre-Motor and Supplementary Motor Cortex	6	73.07 (3.2)	11, 12	23,24
**Inferior parietal cortex**	**39, 40**	**88.29 (3.3)**	**25, 27, 28, 30**	**33, 35, 36, 38**
Primary Somatosensory Cortex	1,2,3	58.15 (3.4)	26	32
Somatosensory Association Cortex	5,7	76.41 (4.5)	29, 31	34,37

The values in parentheses indicate the standard error. Bold values show information about regions of interest.

**Table 2 brainsci-12-00349-t002:** Summary of 2 × 2 × 2 repeated-measurement ANOVAs in each brain region.

	VLPFC			DLPFC			IPC		
	F	*p*	η^2^_p_	F	*p*	η^2^_p_	F	*p*	η^2^_p_
Task type	**6.387**	**0.018 ***	**0.191**	4.185	0.051	0.134	0.100	0.754	0.004
Distractor	**10.525**	**0.003 ****	**0.280**	1.809	0.190	0.063	**4.278**	**0.048 ***	**0.137**
Hemisphere	0.600	0.445	0.022	0.315	0.579	0.012	1.570	0.221	0.055
Task type × distractor	**4.643**	**0.040 ***	**0.147**	1.698	0.204	0.059	**6.008**	**0.021 ***	**0.182**
Task type × hemisphere	0.973	0.333	0.035	0.070	0.794	0.003	0.144	0.708	0.005
Distractor ×hemisphere	1.237	0.276	0.044	0.423	0.521	0.015	0.010	0.919	0.000
Task type × distractor × hemisphere	1.126	0.298	0.040	0.720	0.404	0.026	0.017	0.898	0.001

Bold font represents statistical significance at the *p* < 0.05. VLPFC, ventrolateral prefrontal cortex; DLPFC, dorsolateral prefrontal cortex; IPC, inferior parietal cortex. * *p* < 0.05, ** *p* < 0.01.

## Data Availability

The datasets generated and/or analyzed during the current study are available from the corresponding author upon reasonable request.

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
