# Peer review of "Effect of Audiovisual Cross-Modal Conflict during Working Memory Tasks: A Near-Infrared Spectroscopy Study"

_brainsci, 2022, doi:10.3390/brainsci12030349_

Round 1

Reviewer 1 Report

Why use a bandpass filter with a frequency range between 0.01 Hz and 0.3 Hz?

Previous studies have shown that there are fluctuations with low frequency (LF) at the range 0.1 Hz (0.08–0.12 Hz) in the fNIRS signals called Mayer’s waves. Some studies have used the bandpass filter to reduce physiological noise in the frequency range of 0.01 to 0.1 Hz.

Reviewer 2 Report

Review Brain science

Ms: brainsci-1563986

The authors examined visual-auditory cross-modal conflicts that cause semantic competition during working memory tasks through behavioral and brain activities. To this end 31 young healthy male adults were tested. These participants performed either the PASAT or PVSAT as a target task in combination with the other test as a distraction or without any distraction. In addition to performance, the oxygenated hemoglobin (Oxy-Hb) concentration changes in the frontoparietal regions were measured when performing the tasks. Overall, performance was better in the PVSAT compared to the PASAT. Interestingly, only the PASAT suffered in the condition with distraction compered to without distraction, while performance remained comparable in the visual task PVSAT. The Oxy-Hb concentration changed only in the PASAT and not in the PVSAT. These changes occurred in the bilateral VLPFC and IPC but not in the bilateral DLPFC, which are relevant for the inhibition of visual distraction.

Overall, this study is very interesting and well designed. More specifically the theoretical introduction is thorough and easy to read and follow. Relevant studies are introduced and discussed.

The method is very easy to follow, notably because of the very clear and useful figures that where added, and are well structured, too.

One critical issue is the behavioral analyses: an analysis for RT is missing and should be added. On the one hand since it is an interesting additional measure and to control for a possible speed-accuracy tradeoff.

On a positive note, Table 2 is of great help to have a global overview over the data. However, in the main text, I was a bit lost by the sentences beginning with “The simple main effect…” followed by a description of an interaction between two factors.

The graphs are overall very clear and illustrate the results. In figure 5, the lines indicating how significant the differences are, are not really clearly situated for me (to tilt the line a bit would help, and using framing lines with clearer ends and beginnings).

The discussion is not as clearly structured and harder to follow. [My PhD student (assisting this review had the following remark: For example, after it went a bit vack and forth, I was at first a bit confused when they talk (line 491, line 502) about a “decrease of interference” because I could understand it as if there is less interference in the task, and that is linked with an increased activity, which is not what they want to say I think? In fact, now I am not sure how to understand it anymore, I think they want to say that increased activity is associated with a better handling of interference at the behavioural level? But this seemed a bit odd to me, since we are explicitly in the “Brain activity” sub-section]

The conclusion is overall clear and well summarized.

Minor points:

  • Global structure is very clear, graphs are well chosen and globally clear.
  • 2, line 75: I am not sure I understand the sentence correctly, but I think an “of” might be missing (“in the cross-modal interference control of both visual and auditory”???)
  • line 209: I remains unclear what the baseline is precisely. (although I guess participants must sit in front of the setup and wait for the task to begin?)
  • Line 290: There is no “light blue” frame in the figure 3 (only the other, i.e. green, red, blue).
  • Lines 352-353: the variables may be better distinguished with [] -> [PASAT with distractor-PASAT] and [PVSAT with distractor-PVSAT].
  • Line 366: There is a typo -> “In the VLPC, no significant main effect of the hemisphere was revealed, ….”
  • Line 373: I think they forgot a “without distractor” because in line 372 it reads “with distractor” and it would be probably clearer -> “PASAT without distractor mean, 0.0157±0.0432 [m(mol/l)∗mm]”.
  • Figure 5 (line 380): scales are different in each graph in the vertical axis, would be better keep it consistent.
  • Line 393: in the results for the IPC, they do not talk about the effect of hemisphere. It is not significant cfr table 2, but in the DLPFC it is not significant either and they give the results. Adding it would be better for consistency across the “Brain activity results”.
  • Line 400: “the simple main effect of task type showed…”. However, the main effect was not significant? Or did I misread this? I am curious, because they say there is no main effect of task type (but an interaction distractor*task type).
  • Line 413: Figure 6: add a title above the graduation bar to indicate it is the correlation between the % of correct response and the level of activity in different brain areas, would help to a quicker comprehension.
  • Line 423: Titles of the graphs miss to indicate it is regarding the PASAT test, only the description (line 424) says it.
  • Line 447: not all ROI, the DLPFC had nothing significant?
  • Line 475: Is there a word missing at the beginning of the sentence (“In cross-modal” -> situations? Tasks?)?
